# *N*-Alkylisatin-Loaded Liposomes Target the Urokinase Plasminogen Activator System in Breast Cancer

**DOI:** 10.3390/pharmaceutics12070641

**Published:** 2020-07-07

**Authors:** Lisa Belfiore, Darren N. Saunders, Marie Ranson, Kara L. Vine

**Affiliations:** 1Illawarra Health and Medical Research Institute, Wollongong, NSW 2522, Australia; lisa.belfiore@inventia.life (L.B.); mranson@uow.edu.au (M.R.); 2School of Chemistry and Molecular Bioscience, Faculty of Science Medicine and Health, University of Wollongong, Wollongong, NSW 2522, Australia; 3School of Medical Sciences, University of New South Wales, Sydney, NSW 2052, Australia; d.saunders@unsw.edu.au; 4CONCERT-Translational Cancer Research Centre, Sydney, NSW, Australia

**Keywords:** *N*-alkylisatin, liposome, urokinase plasminogen activator, PAI-2, SerpinB2, breast cancer

## Abstract

The urokinase plasminogen activator and its receptor (uPA/uPAR) are biomarkers for metastasis, especially in triple-negative breast cancer. We prepared anti-mitotic *N*-alkylisatin (*N*-AI)-loaded liposomes functionalized with the uPA/uPAR targeting ligand, plasminogen activator inhibitor type 2 (PAI-2/SerpinB2), and assessed liposome uptake in vitro and in vivo. Receptor-dependent uptake of PAI-2-functionalized liposomes was significantly higher in the uPA/uPAR overexpressing MDA-MB-231 breast cancer cell line relative to the low uPAR/uPAR expressing MCF-7 breast cancer cell line. Furthermore, *N*-AI cytotoxicity was enhanced in a receptor-dependent manner. In vivo, PAI-2 *N*-AI liposomes had a plasma half-life of 5.82 h and showed an increased accumulation at the primary tumor site in an orthotopic MDA-MB-231 BALB/c-Fox1nu/Ausb xenograft mouse model, relative to the non-functionalized liposomes, up to 6 h post-injection. These findings support the further development of *N*-AI-loaded PAI-2-functionalized liposomes for uPA/uPAR-positive breast cancer, especially against triple-negative breast cancer, for which the prognosis is poor and treatment is limited.

## 1. Introduction

Breast cancer is the most common invasive cancer in women worldwide and remains a leading cause of cancer-related morbidity and mortality [1]. While overall survival has improved steadily over the last several decades, breast cancer still accounts for almost half a million deaths each year [2]. Numerous studies and clinical evidence indicate that the urokinase plasminogen activator system (uPAS) play a key role in breast cancer metastasis [3,4]. In this system uPA, secreted as a zymogen, is activated upon binding to its specific cell surface receptor uPAR. Once activated, uPA catalyzes the activation of co-localized plasminogen to plasmin, which in turn directly degrades the components of the extracellular matrix (ECM), promoting further degradation and tissue remodeling, by activating pro-metalloproteinases and activating latent growth factors from the ECM [5]. Localization of the uPAS at the invasive front of tumors thus facilitates cell migration and invasion. In breast cancer, progression-free survival is inversely correlated with uPA and uPAR expression [6,7]. Patients with high uPA mRNA levels are more likely to suffer from metastatic disease [8], and overexpression of uPAR by tumor cells or stromal cells is associated with a poor prognosis for metastatic breast cancer [9]. Amplification and overexpression of uPA and uPAR are thus recognized biomarkers of metastasis and are indicative of an overall poor patient prognosis for breast and several other cancer types [6,7,10,11].

In the triple-negative breast cancer (TNBC) subtype, uPAR was also shown to increase the malignant potential [12] and was identified as a possible novel target for treatment [13,14]. As this subtype lacks the three main breast cancer molecular biomarkers (estrogen receptor (ER), progesterone receptor (PR), and human epidermal growth factor receptor 2 (HER2)), there are no targeted therapy options for TNBC, and the use of conventional chemotherapies remains the standard of care. In addition, TNBC is a markedly heterogeneous subtype that can make treatment problematic. The prognosis of TNBC is generally poor, with high rates of disease recurrence and relapse. The progression-free survival and overall survival rates of TNBC patients are significantly shorter than those of non-TNBC patients [15].

Given the role of uPAS in the promotion of metastasis, targeting uPA/uPAR is a promising therapeutic strategy for TNBC [16,17]. uPA is efficiently and specifically inhibited by the serpin plasminogen activator inhibitor-2 (PAI-2/SerpinB2), forming a covalent complex with uPA/uPAR, which is rapidly internalized via endocytosis receptors [18,19]. Previous work by our group showed that PAI-2 could be used as a targeting ligand for the intracellular delivery of covalently attached cytotoxins to uPAR-positive tumor cells [5,20,21]. Specifically, PAI-2 was conjugated to an *N*-alkylisatin (*N*-AI)-based cytotoxin, a potent microtubule destabilizing agent [22], which could evade P-glycoprotein (P-gp)-mediated efflux in multi-drug resistant cancer cell lines [23,24]. The *N*-AI-PAI-2 conjugate was efficacious in vivo, reducing MDA-MB-231 tumor growth at 1/20th of the dose of free *N*-AI [25]. Given the potency and previous validation of *N*-AI as a cytotoxin for use in anticancer applications, *N*-AI is a promising candidate for further development in drug delivery research. As a hydrophobic molecule, *N*-AI has a low aqueous solubility that limits the amount of drug that can be administered intravenously [26]. However, *N*-AI is amenable to encapsulation within liposomes, in order to improve solubility and physicochemical stability.

Liposomes emerged as a useful delivery system for the transport of drugs and other molecules to solid tumors. Their unique structure allows for the encapsulation of hydrophobic or hydrophilic drugs in the lipid bilayer or aqueous core, respectively [27]. Encapsulated drugs can then be delivered to target cells for intracellular drug release and anti-tumor effect. The inclusion of hydrophilic polymers, most commonly PEG, at the outer surface of the liposome can increase the in vivo circulation time, by reducing recognition and clearance by the MPS [28]. For this reason, PEGylated liposomes were long considered a clinically useful nanoparticle for drug delivery applications. In addition to their versatile drug encapsulation capabilities, liposomes permit the active targeting of specific cell types via the conjugation of ligands to the liposome surface, for drug delivery to cells expressing the target surface receptor(s) of interest [29,30]. Following liposome extravasation into the tumor interstitial space, subsequent ligand-directed surface binding and internalization promotes liposome and drug entry into specific cell types. As targeted liposome formulations combine both passive and active drug delivery mechanisms, ligand-directed liposomes should show superior drug delivery, compared to non-ligand liposomes, depending on the tumor type [31].

Herein, we describe the preparation and characterization of novel *N*-AI-loaded liposomes surface-functionalized with PAI-2 as a ligand for targeting uPA/uPAR. We further evaluated the cellular targeting and in vivo biological properties of these liposomes, using relevant models of uPAR-positive TNBC breast cancer.

## 2. Materials and Methods

### 2.1. Liposome Preparation and Characterization

Liposomes were prepared using the thin-film hydration method [31] and were composed of 20 mM soy PC (L-α-phosphatidylcholine) and 0.6 mM mPEG2000-DSPE (1,2-distearoyl-sn-glycero-3-phosphoethanolamine-*N*-[(polyethylene glycol)-2000]) (Avanti Polar Lipids, AL, USA), with the addition of either 5 mM cholesterol (Sigma-Aldrich, MO, USA) to form empty liposomes, or 5 mM 5,7-dibromo-*N*-(*p*-hydroxymethylbenzyl)isatin (*N*-AI) (prepared in-house [23]) to form *N*-AI-loaded liposomes. *N*-AI encapsulation efficiency was determined by high-performance liquid chromatography (HPLC). Here, *N*-AI-loaded liposomes were mixed with water/acetonitrile (60:40 *v*/*v*) and centrifuged. The *N*-AI concentration was determined using an Atlantis T3 reverse-phase C18 analytical column (Waters, UK) and a Waters HPLC machine (Waters, MA, USA). Analysis was performed using an injected volume of 10 µL, with a gradient elution and monitored with a photodiode array at 435 nm. Concentration was determined by interpolating from a standard curve after analysis of standards and samples using Empower Pro V2 software (Waters, UK).

Liposomes were surface-functionalized with plasminogen activator inhibitor type 2 (PAI-2), as described by our group previously [31]. Unbound PAI-2 was removed from liposomes by size-exclusion chromatography (SEC), using Sepharose CL-4B (Sigma-Aldrich, MO, USA). Western blotting and fluorogenic uPA activity assays were used to detect and quantify PAI-2 conjugated to liposomes and the activity of PAI-2-functionalized liposomes [32]. For Western blotting, PAI-2 in SDS–PAGE gels (reducing conditions) were transferred to PVDF membranes using Bio-Rad transfer equipment (Bio-Rad Laboratories, CA, USA) at 100 V for 1.5 h. Membranes were rinsed in TBST (1× TBS buffer with 0.05% *v*/*v* Tween-20) and blocked using 10% skim milk in TBST for 1 h at RT. After the rinsing membranes were incubated with primary antibody (anti-SerpinB2; Abcam, Cambridge, UK) at 1:2000 dilution in 2% skim milk/TBST at 4 °C overnight. Membranes were washed with TBST four times (10 min each wash) and then incubated with secondary antibody (anti-rabbit-HRP; Abcam, Cambridge, UK) at 1:5000 dilution in 2% skim milk/TBST for 2 h at RT. Membranes were then washed in TBST, three times, for 5 min and then in TBS (no Tween-20), three times, for 5 min. Membranes were developed using ECL peroxidase reaction (Pierce PicoWest ECL reagent; Thermo Fisher Scientific, MA, USA), according to the manufacturer’s instructions. Membranes were visualized using x-ray film, after developing and fixing (Bio-Rad Laboratories, CA, USA) or using a Gel Logic 2200 Digital Imager (Carestream Molecular Imaging, CT, USA). Band intensities were quantified using ImageJ (National Institutes of Health, MD, USA).

### 2.2. Cell Lines and uPA and uPAR Expression

The human mammary epithelial invasive ductal carcinoma cell lines MCF-7 and MDA-MB-231 were purchased from the American Type Culture Collection (ATCC, VA, USA). Cells were cultured in RPMI-1640 medium (Life Technologies, CA, USA) containing 24 mM NaHCO_3_ and supplemented with 10% (*v*/*v*) heat-inactivated fetal bovine serum (FBS; Thermo Fisher Scientific, MA, USA). Cells were maintained in culture at 37 °C in a 95% humidified atmosphere with 5% CO_2_ in a HERAcell incubator (Kendro Laboratory Products, Germany). For passaging, the cells were harvested by treatment with 0.05% trypsin-EDTA (Life Technologies, CA, USA), followed by centrifugation at 300× *g* for 5 min. For the experiments, the cells were harvested by treatment with PBS containing 5 mM EDTA (pH 7.4), followed by centrifugation at 300× *g* for 5 min. Viable cells were counted with a hemocytometer using the Trypan Blue (Sigma-Aldrich, MO, USA) exclusion method. Cell lines were routinely tested and confirmed to be negative for mycoplasma contamination (in-house testing conducted by the Illawarra Health and Medical Research Institute Technical Services Unit). Cell lines were confirmed to be negative for cross-contamination by short-tandem repeat (STR) sequencing (performed by the Garvan Institute of Medical Research, Darlinghurst, Australia).

Expression of uPA and uPAR on the surface of cells was determined by flow cytometry, as described in Appendix A.

### 2.3. Assessment of Cellular Uptake and Localization of Liposomes

MCF-7 and MDA-MB-231 cells were used to assess the cellular uptake of liposomes through flow cytometry and cellular localization by confocal microscopy. For flow cytometry, MCF-7 and MDA-MB-231 cells (2 × 10^5^ cells/well) were seeded into 12-well plates and allowed to attach for 24 h at 37 °C. Liposomes containing 1% (mol/mol) FITC-PEG_2000_-DSPE were added to wells at dilutions ranging from 1:20 to 1:5. At specified time intervals ranging between 15 and 60 min, the supernatant was removed, the cells were washed once with PBS and then harvested using PBS containing 5 mM EDTA (pH 7.4). The cells were then centrifuged (300× *g* for 5 min) and washed three times with PBS, before being resuspended in 200 µL PBS for analysis. The fluorescence intensity was determined by flow cytometry (LSR II flow cytometer; BD Biosciences, CA) (excitation 488 nm, emission collected with 515/20 band-pass filter). FlowJo software (V10; Tree Star Inc., OR, USA) was used to evaluate the mean fluorescence intensity (MFI), to determine the cellular uptake of liposomes.

For confocal microscopy, cells (50,000 per well) were seeded into 8-well µ-Slide chambered coverslips (ibidi, Germany) and incubated for 24 h at 37 °C. Cells were allowed to reach 80% confluence before the addition of liposomes. Liposomes containing 1% or 10% (mole % of liposome phospholipid) FITC-PEG_2000_-DSPE, or 0.625% (mole % of liposome phospholipid) octadecyl rhodamine B chloride (R18; Invitrogen, CA, USA) were added to cells at dilutions ranging from 1:5 to 1:10, and incubated for 30 min to 2 h at 37 °C. The supernatant was removed and the wells were rinsed three times with PBS, before LysoTracker Green DND-26 (excitation/emission 504/511 nm; Thermo Fisher Scientific, MA, USA) was added to each well (50 nM final concentration), immediately prior to imaging. Live imaging of cells in PBS was performed using a Leica TCS SP5 Confocal Microscope (Leica Microsystems, Germany) and the images were acquired using a 63× oil immersion lens. Images were analyzed using the Leica Application Suite software (V10; Leica Microsystems, Germany).

### 2.4. 3D multicellular Tumor Spheroid Cytotoxicity Assays

MCF-7 or MDA-MB-231 cells were seeded into ultra-low attachment 96-well plates (Sigma-Aldrich, MO, USA), at a density ranging between 625 and 5000 cells per well and incubated at 37 °C to promote spheroid formation. Liposomes were serially diluted in PBS and incubated with cells for up to 96 h (each concentration tested in triplicate).

### 2.5. Pharmacokinetics and Biodistribution of N-AI PAI-2 Liposomes in Mice

Female BALB/c-Fox1nu/Ausb nude immunocompromized mice (5 weeks old) (Australian BioResources, Moss Vale) were housed in isolator cages at the University of Wollongong animal facility. Mice were given food and water ad libitum and kept on a 12-h light/dark cycle for the duration of the experiment. Mice were allowed to acclimatize for 2 weeks before commencement of the experiment. All experiments were conducted in accordance with the ‘NHMRC Australian Code for the Care and Use of Animals for Scientific Purposes’, which requires 3R compliance (replacement, reduction, and refinement) at all stages of animal care and use, and the approval of the Animal Ethics Committee of the University of Wollongong (Australia) under protocol AE13/18. MDA-MB-231 cells (ATCC; mycoplasma negative and STR profiled) were resuspended in PBS (no Ca/Mg; pH 7.4; Sigma-Aldrich, MO, USA) and counted using Trypan blue (Sigma-Aldrich, MO, USA) and a hemocytometer. Insulin syringe needles (29-gauge; BD Biosciences, NJ, USA) were used to inject 50 µL of cell suspension (containing 2 × 10^6^ cells) into the upper left mammary fat pad. Mice were injected one cage at a time and the injection order of cages was randomized. Mice were monitored closely following the injection of cells, and the tumors were observed to form at approximately 3 weeks post-injection.

Tumors were < 100 mm^3^ upon commencement of liposome treatment. Mice were randomly allocated to treatment (*N*-AI liposome or *N*-AI PAI-2 liposome) and time-point (10 min, 3 h, 6 h, 24 h, 48 h or 96 h) groups (4 mice per cohort). Treatments (100 µL; 4 µCi/mouse) were administered intravenously via a single lateral tail-vein injection. Mice that were deemed significantly (±10%) smaller or larger in weight than their cage mates had their dose volume adjusted proportionally, based on their weight, relative to the average of their cage mates. The ^3^H-CHE radioactivity (liposome) in the plasma, kidneys, liver, spleen, lungs, tumor and tail (for injection correction) was quantified using previously published methods [33]; further details are provided in Appendix A.

### 2.6. Toxicology of N-AI Liposomes in Mice

The toxicology of *N*-AI liposomes was determined in female BALB/c mice via single or multiple lateral tail-vein injections. Details are described in Appendix A.

### 2.7. Data Analysis

All data analysis, including the generation of graphs and statistical tests, was performed using GraphPad Prism version 7 for Windows (GraphPad Software, CA, USA), unless stated otherwise. Data are presented as the mean ± standard deviation (s.d.) or standard error of the mean (s.e.m.) as stated. Pairwise comparisons were made using Student’s *t*-test and multiple comparisons were made using one-way ANOVA with Tukey’s post-test.

## 3. Results

### 3.1. Preparation and Characterization of Liposomes

Modifications to our previously reported method [33] were used to prepare and characterize the PEGylated liposomes containing the potent microtubule-destabilizing agent 5,7-dibromo-*N*-(*p*-hydroxymethylbenzyl)isatin (*N*-AI; see Appendix A for chemical structure), surface functionalized ± PAI-2. The particle diameter, polydispersity index (PDI), peak intensity, and zeta potential for all liposome preparations are summarized in Table 1.

The size and morphology of *N*-AI liposomes were further confirmed by measurement of liposome diameter from cryogenic transmission electron microscopy (cryo-TEM) images (Figure 1a). The average diameter of *N*-AI liposomes (138.7 ± 18.4 nm; Figure 1b) was similar to that determined using dynamic light scattering (139.9 ± 3.9 nm; Table 1). Cryo-TEM additionally revealed *N*-AI liposomes to be spherical, monodisperse, and unilamellar. The concentration of *N*-AI encapsulated in the liposomes could not be determined by spectrophotometry, as the liposome phospholipid interfered with the peak absorbance of *N*-AI at 310 nm and 435 nm (Appendix A). Therefore, the concentration of *N*-AI loaded into liposomes was determined by HPLC, which revealed an *N*-AI concentration of 2.2 mM, equating to a 43.1% entrapment efficiency based on the starting amount of *N*-AI used in the liposome preparation (Appendix A). This translated into 7.3% *w*/*w*
*N*-AI loaded per unit weight of the soy PC, indicating the percentage of mass of the liposome that is due to the encapsulated drug.

PAI-2 was incubated with preformed liposomes containing mal-PEG_2000_-DSPE, to allow conjugation to the liposome surface. Unconjugated PAI-2 was removed using size-exclusion chromatography (SEC; Figure 1c). Analysis of fractions by spectrophotometry revealed that the unconjugated PAI-2 (peak 2) had separated from the covalently attached PAI-2 on the liposome surface (peak 1). As PAI-2 could not be detected or quantified using commercial biochemical protein assays, due to phospholipid interference [34] (data not shown), Western blotting was used to confirm successful conjugation of PAI-2 to liposomes (Figure 1d). Covalent conjugation of PAI-2 to liposome phospholipid (PEG-DSPE; molecular weight ~2940 kDa) to form PAI-2-PEG-DSPE was confirmed by a lag in gel migration of PAI-2 in the peak 1 fraction (sample #1), relative to the peak 2 fraction (sample #2), which corresponded to the 45 kDa molecular weight of free PAI-2. The amount of PAI-2 associated with the liposome fraction in sample #1 was 42 ng, after interpolation from a standard curve.

An important step in characterizing ligand-functionalized liposomes was to confirm whether the targeting ligand(s) retain activity against the target receptor once bound to the liposome surface. The uPA inhibitory activity of PAI-2-liposomes was assessed using enzymatic assays. A significant reduction in the rate of FLU was observed for the EMP PAI-2 liposomes (43.5 ± 24.9 FLU/min) compared to the EMP liposomes (4026.9 ± 206.2 FLU/min) (Figure 1e). EMP PAI-2 liposomes were as effective at inhibiting uPA activity, as the unconjugated PAI-2 (95–100% inhibition) demonstrating that PAI-2 liposomes were fully active.

### 3.2. PAI-2 Liposomes Are Taken up by Cells through RME-Dependent and Non-Dependent Mechanisms

Prior to assessing cellular uptake, the MCF-7 and MDA-MB-231 breast cancer cell lines were profiled for cell surface uPA and uPAR expression through flow cytometry. MDA-MB-231 cells showed a significantly (P < 0.001) higher mean fluorescent intensity for uPAR and uPA (MFI; 11.82 ± 0.90 and 6.66 ± 0.97, respectively) than MCF-7 cells (MFI; 0.26 ± 0.03 and 2.49 ± 0.10, respectively; Figure 2a).

The cellular uptake of liposomes was determined by flow cytometry, using FITC-PEG-DSPE incorporated into the liposome bilayer. PAI-2-functionalized (EMP PAI-2; 152.6 ± 8.7 nm) and non-functionalized (EMP; 152.8 ± 11.7 nm) FITC liposomes were incubated with MCF-7 cells (low uPA/uPAR) and MDA-MB-231 cells (high uPA/uPAR) for 45 min. A significant increase in the EMP PAI-2 liposome uptake was observed in the MDA-MB-231 cells at 5 mM and 2.5 mM liposome concentrations (*p* < 0.0001 and *p* < 0.001, respectively) relative to the EMP liposomes, but not at 1.25 mM liposome concentration. No significant differences were observed between the uptake of EMP and EMP PAI-2 liposomes in the MCF-7 cells, at any liposome concentrations (*p* > 0.05; Figure 2b).

For the cellular localization of EMP PAI-2 liposomes through confocal microscopy, the intensely fluorescent fluorophore R18 was used to label liposomes. Dynamic light scattering revealed average diameters of 131.3 ± 2.5 nm and 131.2 ± 6.6 nm for EMP and EMP PAI-2 R18-labelled liposomes, respectively. A strong fluorescent signal from R18-labelled liposomes was detected at the cell membrane, within the cytoplasm and within lysosomes (indicated by colocalization of liposome and LysoTracker), 1 h post-incubation for both cell lines (Figure 2c), indicating cellular uptake for both EMP PAI-2 liposomes and EMP liposomes.

### 3.3. Cytotoxicity of N-AI PAI-2 Liposomes against Breast Cancer Cells

Treatment of MCF-7 and MDA-MB-231 cells for 72 h with *N*-AI PAI-2 liposomes but not EMP PAI-2 liposomes (at an equivalent phospholipid concentration), resulted in a dose-dependent decrease in cell viability for both cell lines, consistent with intracellular delivery of the cytotoxic *N*-AI (Appendix A). The cytotoxic effect of *N*-AI PAI-2 liposomes against MDA-MB-231 cells (IC_50_ of 5.40 ± 1.14 µM) was significantly greater (*p* < 0.01) than the MCF-7 cells (IC_50_ of 31.84 ± 8.20 µM). EMP PAI-2 liposomes elicited some degree of cytotoxicity in both cell lines, at the highest liposome concentrations tested.

### 3.4. Cytotoxicity of N-AI PAI-2 Liposomes against Breast Cancer Spheroids

MCF-7 and MDA-MB-231 cells are reported to form spheroids under low-attachment growth conditions [35]. EMP PAI-2 and *N*-AI PAI-2 liposomes at equivalent phospholipid concentrations were incubated with the preformed spheroids and imaged every 24 h (Figure 3). Spheroids treated with EMP PAI-2 liposomes showed continued growth and an increase in spheroid diameter, over time. In contrast, treatment with *N*-AI PAI-2 liposomes showed a time- and concentration-dependent disassembly of the spheroid structure, at concentrations above 62 µM for both cell lines (Figure 3a). However, MDA-MB-231 spheroids appeared to be more sensitive to *N*-AI PAI-2 liposome treatment, which showed clear evidence of spheroid dissociation, as early as 24 h, compared to the MCF-7 spheroids (Figure 3b). By 48 h, the MDA-MB-231 spheroids were almost completely dissociated in contrast to MCF-7 spheroids. A comparison of the Calcein AM stained spheroids after 96 h found MDA-MB-231 spheroids treated with *N*-AI and *N*-AI PAI-2 to be fully dissociated, while the MCF-7 spheroids, although smaller than the control (EMP and EMP PAI-2), remained largely intact (Figure 3c).

### 3.5. Pharmacokinetics and Biodistribution of N-AI PAI-2 Liposomes

To determine the pharmacokinetic and organ distribution profiles of *N*-AI liposomes and *N*-AI PAI-2 liposomes in tumor-bearing mice, liposomes were labelled with tritiated cholesteryl hexadecyl ether (^3^H-CHE), to enable their detection in plasma and tissues, through liquid scintillation counting. Liposomes were monodispersed with average diameters of 115 ± 34 nm and 117 ± 39 nm, for *N*-AI and *N*-AI PAI-2 liposomes, respectively. Scintillation counts of the two liposome stock preparations were 319,698 CPM/ml and 312,163 CPM/ml for *N*-AI and *N*-AI PAI-2 liposomes, respectively. The plasma half-life was determined to be 5.63 h and 5.82 h for the *N*-AI and *N*-AI PAI-2 liposomes, respectively (Figure 4a). The plasma clearance profiles of the two liposomes and the pharmacokinetic parameters from the curve-fitting analysis were not significantly different (*p* > 0.05) (Table 2).

At 24 h, 48 h, and 96 h, tumor uptake of the *N*-AI and *N*-AI PAI-2 liposomes was not significantly different (*p* > 0.05). Liposome accumulation in the kidneys, liver, spleen, and lungs at each time-point was similar between the *N*-AI and *N*-AI PAI-2 liposomes (Figure 4c–f). The trends indicated increased clearance via the liver and spleen, over time, with clearance via the kidneys and accumulation in the lungs was minimal for both liposome formulations (Figure 4g–h).

Tumors were removed from mice and analyzed for tritiated liposome signal. The results showed rapid accumulation of *N*-AI PAI-2 liposome signal in tumors, compared to *N*-AI liposomes, as indicated by the significantly increased %ID at 10 min, 3 h, and 6 h post-injection (*p* < 0.001; Figure 4b).

### 3.6. Maximum Tolerated Dose of N-AI-Loaded Liposomes in Mice

*N*-AI liposomes were well-tolerated in mice, when given as an intravenous (i.v.) single bolus dose up to 20 mg/kg *N*-AI or multiple fractionated dose, up to 100 mg/kg *N*-AI. This exceeded that of free *N*-AI, which had a maximum tolerated single bolus dose of 8.5 mg/kg (Appendix A). Mice treated with free drug at concentrations above 8.5 mg/kg, as well as the equivalent volume of the DMSO/PBS vehicle without drug, displayed adverse signs of intolerance immediately upon injection, including lethargy, hind-leg paralysis, tremors, and difficulty breathing [36,37], preventing higher concentrations of free *N*-AI from being tested. Liposome encapsulation therefore provided an injectable formulation of *N*-AI, with improved tolerability than the free drug.

## 4. Discussion

The uPA system is recognized to play a central role in the ability of breast cancer cells to escape the primary site of the tumor and colonize other parts of the body. Hence, targeting this system using novel approaches might be effective in the treatment of highly invasive or metastatic breast cancer. This work aimed to improve upon the solubility and delivery of the potent *N*-alkylisatin (*N*-AI) cytotoxin to uPA/uPAR positive breast cancer cells, through the conjugation of PAI-2 to the surface of PEGylated *N*-AI-loaded liposomes.

The successful encapsulation of a hydrophobic drug into liposomes can greatly enhance the aqueous solubility and bioavailability of the molecule, and therefore increase the suitability for its use in parenteral drug delivery applications. With a logarithmic octanol/water partition coefficient (LogP) of 3.39, the aqueous solubility of non-liposomal *N*-AI is negligible. In this work, the thin film hydration method was successfully used to load *N*-AI into the bilayer of soy PC PEGylated liposomes, to achieve a final *N*-AI concentration of 2.2 mM in aqueous solution. *N*-AI was substituted for cholesterol in the formulation, in order to increase the drug-loading capacity of *N*-AI in the bilayer. As the molecular weight of *N*-AI and cholesterol were similar, and both were hydrophobic molecules, we achieved a greater encapsulation of *N*-AI in the liposomes, in the absence of cholesterol, without affecting the zeta potential, liposome size, or stability (Appendix A).

The zeta potential of liposomes is dependent on a number of factors [38], while PEG itself does not affect the surface charge of liposomes, PEG-DSPE introduces a negative surface potential due to the phosphate diester moiety [39]. In this study, liposomes displayed a slightly negative, but near-neutral, zeta potential, which did not vary greatly with *N*-AI encapsulation or PAI-2 conjugation to the liposome surface. Although it was noted that PAI-2-functionalized liposomes were slightly more negative than the non-functionalized liposomes, as PAI-2 has a predicted isoelectric point of 5.4 and therefore a negative charge at physiological pH [40]. It was reported that liposomes with mildly charged or near-neutral surfaces show a propensity to aggregate faster than liposomes with a strong surface charge, as the latter show a greater particle–particle repulsion and hence are more electrostatically stabilized in suspension [41]. However, our liposome formulations were found to be remarkably stable with no aggregation detected or drug leakage under the storage conditions for >4 weeks (data not shown).

The optimization of surface-functionalized liposomes through the covalent attachment of targeting moieties to the outer lipid leaflet is complex. According to a recent study by Herda et al., only 3.5% of proteins attached to the surface of SiO_2_−PEG_8_−Tf particles had appropriate orientation for receptor recognition [42]. Furthermore, increasing antibody density on the surface of nanoparticles was found to reduce receptor-dependent targeting [43]. In this study, the conjugation of PAI-2 to the surface of liposomes was confirmed by Western blotting and the maintenance of PAI-2 inhibitory activity post-conjugation was validated by uPA assay. PAI-2-liposomes were as effective at inhibiting the enzymatic activity of uPA as the unconjugated PAI-2, demonstrating that PAI-2 liposomes were fully active. While Western Blotting was successful in qualitatively confirming the conjugation of PAI-2 to liposomes and in confirming the absence of unconjugated PAI-2 in the purified liposome, the use of emerging methods such as single molecule fluorescent imaging to quantify ligand density on the liposome surface would enable a more precise characterization of targeted liposome formulations [44].

As reported previously, cell surface uPA and uPAR expression is low in MCF-7 cells relative to the MDA-MB-231 cells [12,45]. This difference in uPAR expression was associated with a significant increase in fluorescently labelled EMP PAI-2 liposome uptake, relative to the non-functionalized EMP liposomes by MDA-MB-231 cells, but not by MCF-7 cells. Liposomes can be taken into cells via several different mechanisms, including adsorption, lipid exchange, intracellular membrane fusion, and receptor-mediated endocytosis (RME) [46]. The presence of a fluorescent signal from cells treated with EMP fluorescently labelled liposomes indicates that these liposomes were taken up into cells by fusion or other non-specific mechanisms, rather than by RME. In contrast, the uptake of EMP PAI-2 liposomes by the MDA-MB-231 cells was greater than the uptake of EMP liposomes in MDA-MB-213 cells. As the average liposome diameters of the FITC-labelled EMP and EMP PAI-2 liposomes were equivalent, this difference in uptake was likely due to the presence of PAI-2 at the liposome surface and interaction with uPA/uPAR overexpressed on the surface of MDA-MB-231 cells. Competition binding studies using excess PAI-2 or uPAR antibody could be used to further confirm this mechanism [47,48], however, the uPA/uPAR-dependent uptake of PAI-2 has been extensively characterized by our group [18,19,20,49] and others [50]. Studies assessing the colocalization of liposome signal with lysosomes through confocal microscopy, further confirmed that liposomes were indeed internalized by cells and accumulated in lysosomes, in addition to being present elsewhere in the cell. This result was not unexpected, given that the liposomes were incubated with cells at a high phospholipid concentration, creating an environment where liposomes in solution would passively fuse with cell membranes over time [46]. Collectively, our results indicate that EMP and EMP PAI-2 liposomes are taken into cells via a number of mechanisms, including RME.

In the multicellular tumor spheroid experiments, imaging of spheroids over 96 h indicated a concentration- and time-dependent destruction of both MCF-7 and MDA-MB-231 spheroids, when treated with *N*-AI PAI-2 liposomes. However, MDA-MB-231 spheroids treated with *N*-AI PAI-2 liposomes showed a greater destruction of the spheroid architecture than MCF-7 spheroids at lower *N*-AI PAI-2 liposome concentrations and at earlier time-points. We posit that this effect was receptor-dependent, due to the uPA/uPAR targeting of PAI-2 functionalized liposomes, but acknowledge that it might also be the result of differences in cellular adhesion molecules involved in spheroid formation and cell–cell interactions between the two cell lines [51], thereby influencing liposome perfusion. While spheroid models are increasingly used to study anti-tumor drug effects, a major limitation is that they mimic only the avascular region of in vivo tumors, and exclude important aspects of tumorigenesis, such as the vasculature, immune system, and fluid dynamics. This is especially important in the context of evaluating the efficacy of nanomedicines, where the enhanced permeability and retention (EPR) effect is likely at play and vascular permeability is a relevant factor in their accumulation at the tumor site.

Determining the in vivo properties of novel nanotherapies is important for evaluating how a new nanoparticle formulation can be expected to perform in humans. In this work, the pharmacokinetics and tissue distribution of *N*-AI and *N*-AI PAI-2 liposomes were evaluated in an orthotropic MDA-MB-231 breast tumor xenograft mouse model. The addition of PAI-2 to the surface of *N*-AI-loaded liposomes did not significantly alter the in vivo blood clearance properties of the formulation, but did increase accumulation of liposomes at the primary tumor site relative to non-functionalized liposomes. The PEGylated liposome formulation contained 10 mol% PEG-DSPE and ranged between 130 nm and 150 nm in diameter. This liposome size range and PEG density has been reported to avoid rapid clearance by the mononuclear phagocyte system (MPS) in circulation, and to utilize the EPR effect to extravasate and accumulate at the site of tumors for drug delivery [42]. The accumulation of nanoparticles in tumors via EPR is dependent on a number of factors, including interstitial fluid pressure, vascularity of the tumor, and the in vivo circulation time of the nanoparticle formulation [52]. As the plasma half-lives of the two liposomes were largely equivalent, the presence of PAI-2 at the liposome surface might have affected liposome extravasation and uptake at the tumor site, with PAI-2 liposomes binding to uPAR expressed by tumor cells, as was observed in the in vitro experiments. The difference in uptake might also be due to slight differences in surface charge. Surface charge was previously shown to affect tumor uptake of nanoparticles [48], whereby histological analysis showed that negatively charged and neutral liposomes are able to extravasate at the site of the tumor, while positively charged liposomes remain associated with the vascular endothelium, limiting their suitability for tumor-targeting applications [53].

Finally, the maximum tumor accumulation of the liposomes reported in this study was 0.5% of the ID at the 10 min time-point for *N*-AI PAI-2 liposomes and 0.02% for *N*-AI liposomes. These values were comparable to other PEGylated nanoparticles, which typically showed 1% or less of the total ID reaching the site of the primary tumor. Notably, the high tumor accumulation of Doxil^®^ in humans (reported as high as 10% of the ID) was due in large part to the very long circulation half-life (up to 45 h) of the formulation [54]. Given that the in vivo half-life of *N*-AI PAI-2 liposomes was 5.82 h, modifications to the ‘stealth’ coating of liposomes to promote increased blood circulation and reduce MPS clearance, might further improve liposome uptake into tumors. Following systemic administration, the surface of PEGylated liposomes is modified by protein adsorption, forming a protein corona. Future studies should also aim to characterize the composition of the protein corona, before conducting in vivo efficacy studies, in order to further understand the mechanism of tumor uptake.

## 5. Conclusions

This work supports the rationale for targeting uPAR-positive breast cancer cells, using *N*-AI-loaded PAI-2-functionalized liposomes, and provides a basis for the further development of liposomes that can target heterogeneous tumor cells within the TNBC subtype, in which uPAR was shown to play a key role in driving metastasis. Further research is needed to clarify if and how the potency of *N*-AI as a cytotoxin could be translated into an anti-tumor growth effect by targeting uPAR-positive tumors. The utilization of advanced preclinical models and methods will enable an enhanced evaluation of *N*-AI PAI-2 liposomes in the in vivo context. Future studies assessing the efficacy of *N*-AI PAI-2 liposomes in TNBC are therefore warranted.

## Figures and Tables

**Figure 1 pharmaceutics-12-00641-f001:**
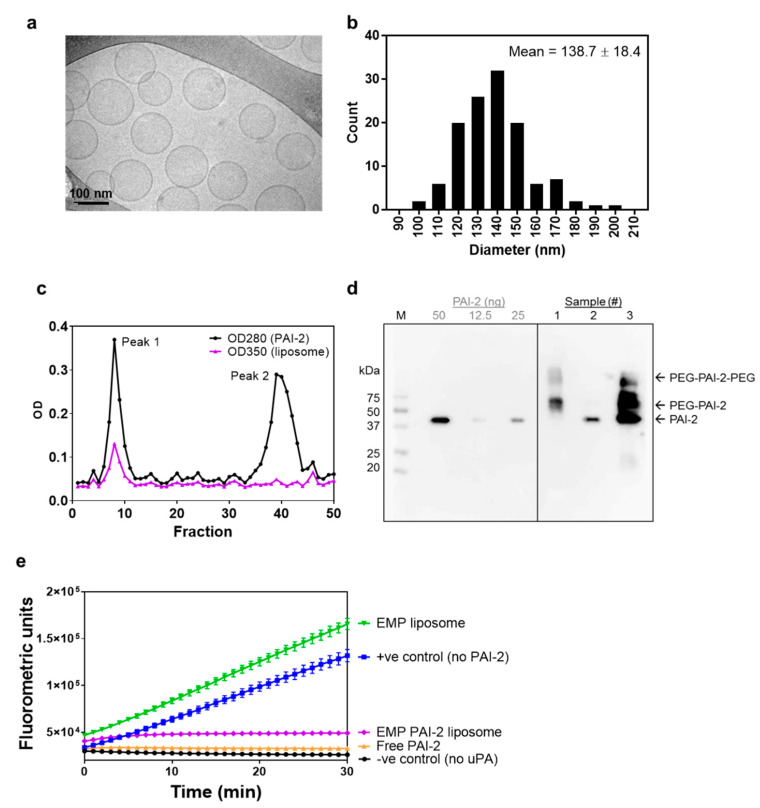
Characterization of *N*-AI-loaded liposomes. (**a**) Representative cryo-TEM image of *N*-AI-loaded liposomes. (**b**) Determination of the average liposome diameter from cryo-TEM image analysis. (**c**) Size-exclusion chromatograph of the PAI-2 liposome fractions after conjugation, including PAI-2 liposomes (peak 1) and unbound PAI-2 (peak 2). (**d**) Western blot detection of PAI-2 in size-exclusion fractions (1, 2), un-purified liposomes (3), and purified PAI-2 (50, 25 and 12.5 ng). OD = optical density, M = marker, PEG = polyethylene glycol. (**e**) Kinetic inhibition curves for unconjugated PAI-2 versus PAI-2 conjugated to empty liposomes (EMP PAI-2), against uPA in solution. Empty liposomes were included as a fluorescence control. Values are means ± s.d. (*n* = 3).

**Figure 2 pharmaceutics-12-00641-f002:**
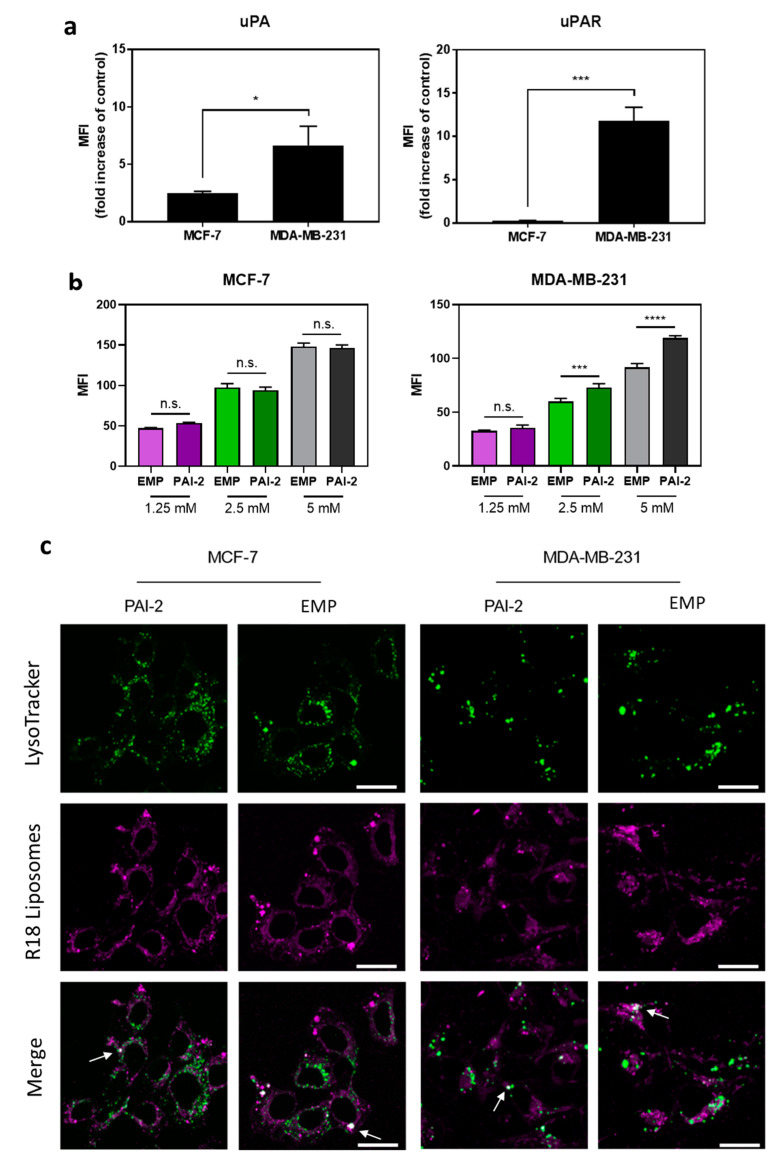
Cell surface expression of uPA/uPAR and cellular uptake of PAI-2 FITC-labelled liposomes by breast cancer cells. (**a**) MCF-7 cells and MDA-MB-231 cells were incubated with antibodies against human urokinase plasminogen activator (uPA), its receptor (uPAR), or an isotype control antibody (IgG) and cell surface expression analyzed by flow cytometry. (**b**) MCF-7 cells (left) and MDA-MB-231 cells (right) were incubated with empty non-functionalized (EMP) FITC liposomes or empty PAI-2-functionalized (PAI-2) FITC liposomes for 45 min, and were analyzed by flow cytometry. MFI = mean fluorescence intensity Data are the mean ± s.d. (*n* = 3). *: *p* < 0.05; ***: *p* < 0.001; ****: *p* < 0.0001; n.s. = not significant (*p* > 0.05). (**c**) EMP liposomes and PAI-2 liposomes were labelled with R18 and incubated with cells at a liposome concentration of 2.5 mM for 1 h. LysoTracker green was added to visualize lysosomes. Arrows point to white foci, which indicate colocalization of green and magenta signals. Representative images are shown. Scale bars are 25 µm.

**Figure 3 pharmaceutics-12-00641-f003:**
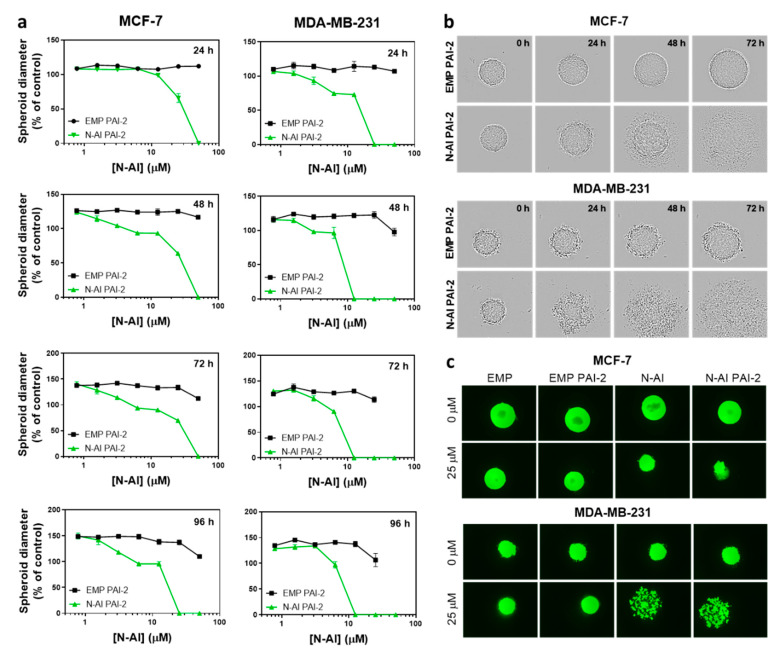
Cytotoxic effect of *N*-AI PAI-2 liposomes on breast cancer spheroids. (**a**) Spheroid diameter was measured after incubation with EMP PAI-2 liposomes or *N*-AI PAI-2 liposomes with MCF-7 and MDA-MB-231 multicellular tumor spheroids, over a period of 96 h. (**b**) Representative bright-field images and (**c**) fluorescent images, following the addition of calcein-AM to visualize the viable cells were captured at the same magnification (*n* = 3). Spheroids in (**b**, **c**) were treated with 25 µM *N*-AI or the equivalent concentration of phospholipid in liposomal formulation. Scale bars are 100 µm. Data are the mean ± s.d. (*n* = 3).

**Figure 4 pharmaceutics-12-00641-f004:**
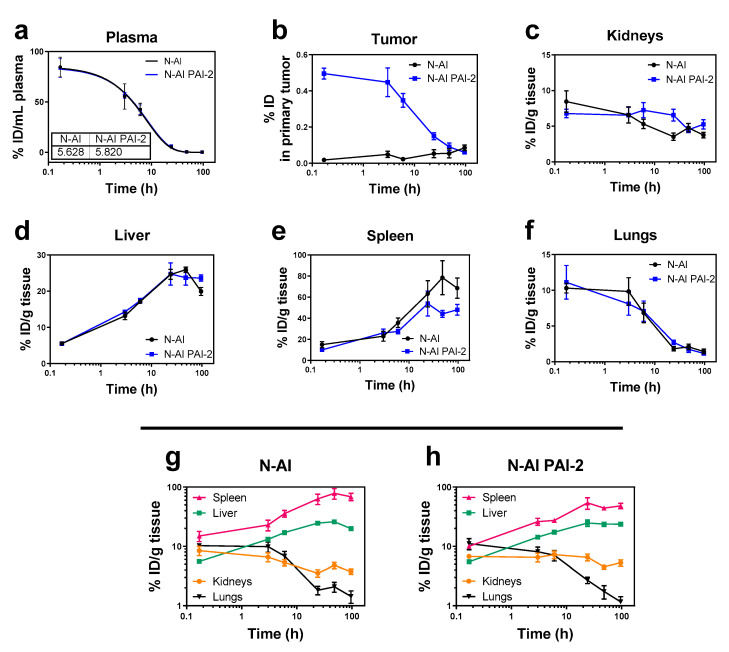
Biodistribution and pharmacokinetics of the radiolabeled liposomes in mice. *N*-AI and *N*-AI PAI-2 liposomes were labeled with tritiated cholesteryl hexadecyl ether (^3^H-CHE) and administered intravenously as a single bolus dose. Tritiated signal was measured in the (**a**) plasma, (**b**) tumor, (**c**) kidneys, (**d**) liver, (**e**) spleen, and (**f**) lungs at each time-point. Kidney, liver, spleen, and lung data are also presented on the same graph for the (**g**) *N*-AI liposomes and (**h**) *N*-AI PAI-2 liposomes, for biodistribution comparison. Results are expressed as the percentage of injected dose (ID) per gram of tissue or milliliter of plasma, and as the percentage of the injected dose (ID) in the whole analyzed primary tumor. Values are the mean ± s.e.m. (*n* = 4).

**Table 1 pharmaceutics-12-00641-t001:** Characterization of empty and *N*-AI PEGylated liposomes. Empty (EMP) liposomes, *N*-AI-loaded (*N*-AI) liposomes, empty PAI-2-functionalized (EMP PAI-2) liposomes, and *N*-AI-loaded PAI-2-functionalized (*N*-AI PAI-2) liposomes were prepared by the thin-film hydration method and analyzed by dynamic light scattering. Values are means ± s.d. (*n* = 3).

Liposome	Diameter (nm)	Polydispersity Index	Peak Intensity %)^#^	Zeta Potential (mV)	Phospholipid (mM)
EMP	137.6 ± 5.6	0.067 ± 0.04	100	−3.63 ± 0.80	16.44
*N*-AI	139.9 ± 3.9	0.093 ± 0.02	100	−3.64 ± 0.59	16.45
EMP PAI-2	139.7 ± 4.9	0.109 ± 0.02	100	−4.05 ± 0.53	16.67
*N*-AI PAI-2	141.1 ± 5.0	0.086 ± 0.03	100	−4.66 ± 0.52	16.62

^#^ Percent of particles present relative to the total particle population.

**Table 2 pharmaceutics-12-00641-t002:** Pharmacokinetic parameters of *N*-AI and *N*-AI PAI-2 liposomes. *N*-AI and *N*-AI PAI-2 liposomes were labeled with tritiated cholesteryl hexadecyl ether (^3^H-CHE) and administered intravenously as a single bolus dose. ^3^H-CHE signal was measured in plasma at each time-point to determine the following parameters.

PK Parameter	*N*-AI	*N*-AI PAI-2
C_max_ (% ID/mL)	84.66 (± 9.79)	83.76 (± 9.25)
K_elim_ α (fast) min^−1^	0.061	0.058
K_elim_ β (slow) min^−1^	0.002	0.002
T_1/2_ α (fast) min	11.419	12.050
T_1/2_ β (slow) min	408.152	410.843
Correlation coefficient (R^2^)	0.9629	0.9836
AUC (% ID/min/mL)	860.3 (± 66.89)	873.4 (± 50.79)

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
