# Peer review of "N-Alkylisatin-Loaded Liposomes Target the Urokinase Plasminogen Activator System in Breast Cancer"

_pharmaceutics, 2020, doi:10.3390/pharmaceutics12070641_

Round 1
Reviewer 1 Report
The manuscript entitled “N-alkylisatin-loaded liposomes target the urokinase plasminogen activator system in breast cancer” by Belfiore et. al. reported the preparation of N-AI loaded and PAI-2 functionalized liposomes for targeting uPA/uPAR. Overall, the manuscript is properly organized, and many studies were well-performed in vitro and in vivo.
One big question is that since PAI-2 liposomes were found to be taken up by cells also in an uPA/uPAR-nondependent manner, how to ensure that liposomes can be selectively taken up by cancer cells? Throughout the manuscript, the authors only use two breast cancer cell lines (MCF-7 and MDA-MB-231) to show that a certain degree of differences exist due to the different levels of uPA/uPAR expressed by the two cell lines. However, the authors didn’t test the selectivity of the liposomes toward cancer cells as compared with normal cells. Without addressing this concern, I don’t think the work will be useful for researchers in this field.
Reviewer 2 Report
The manuscript reports an interesting development study for Nalkilisatin loaded liposomes for active targeting through UPA to breast cancer.
The manuscript has no major criticisms and it is well written and accurate and scientifically sound. therefore, I recommend publication after addressing a few minor questions and issues:
1) The AA have partially discussed these points that however need some more comments.
Why have not the AA evaluated the effect on tumor growth? Data show that the labeled liposomes can accumulate in the tumor but two questions remain unanswered: if the delivered dose is sufficient for antitumor activity and why the liposomes that are supposed to be stealth show a relatively short half-life. Liposomes seem not to have been optimized under this point of view. Perhaps pegylation was not adequate to increase the physiologic halflife of the vesicles. How have the AA chosen the level of DSPE-PEG2000 to add? How have they chosen the level of functionalization?
how the AA explains the lack of PK differences between functionalized and non-functionalized liposomes'
2) From cryo-TEM. The functionalized PEG corona does not seem to be so evident on the liposome surface. Perhaps the level of pegylation is too low. This may correlate with the short half-life.
3) Lines 390-399. Liposomes chemical-physical properties do not change with functionalization. Zeta potential is identical and it is known to drop nearly to zero in stealth liposomes due to PEG which provides not charge but steric stabilization. However, a slight increase of negative charge should be expected in functionalized vesicles, perhaps the lack of an effect may be related to the low level of PAI-2.
Have the AA checked what happens to the liposomes when exposed to whole blood? This may help the interpretation of in vivo data.
4) Is liposome labeling changing the lipid ratio in the formulation? The logical answer should be no.
5) Table 1- Explain better the meaning of the % intensity reported.
6) Avoid multiple abbreviations repeat in figure captions. Try to define abbreviations only once
Reviewer 3 Report
The manuscript by Belfiore et al., systematically describes the development and evaluation of uPAR-targeted liposomes containing N-alkyl isatin. The authors have carried out adequate studies to demonstrate the potential of uPAR targeting and N-alkyl isatin. Please see below my comments
- The authors should include the structure of N-alkyl isatin (NAI) used in this study.
- The authors mention that because of the similar molecular size of NAI used in this study, they did not include cholesterol in the liposomal formulation. Based on the chemical description of NAI [5,7-dibromo-N-(p-hydroxymethylbenzyl)isatin] used in this study, it seems unlikely that it could be a substitute for cholesterol. Cholesterol incorporation imparts sufficient rigidity to the liposomal formulation and allows for long-term stability. Could the authors clarify this point?
- The authors should include additional details about NAI as a potential drug candidate. Is it better compared to taxanes? Please include the relevant details.
Round 2
Reviewer 1 Report
Thanks for the authors' responses.